# RVT: Robotic View Transformer for 3D Object Manipulation

**Ankit Goyal, Jie Xu, Yijie Guo, Valts Blukis, Yu-Wei Chao, Dieter Fox**
NVIDIA

**Abstract:** For 3D object manipulation, methods that build an explicit 3D representation perform better than those relying only on camera images. But using explicit 3D representations like voxels comes at large computing cost, adversely affecting scalability. In this work, we propose RVT, a multi-view transformer for 3D manipulation that is both scalable and accurate. Some key features of RVT are an attention mechanism to aggregate information across views and re-rendering of the camera input from virtual views around the robot workspace. In simulations, we find that a single RVT model works well across 18 RLBench tasks with 249 task variations, achieving 26% higher relative success than the existing state-of-the-art method (PerAct). It also trains 36X faster than PerAct for achieving the same performance and achieves 2.3X the inference speed of PerAct. Further, RVT can perform a variety of manipulation tasks in the real world with just a few (∼10) demonstrations per task. Visual results, code, and trained model are provided at: https://robotic-view-transformer.github.io/.

**Keywords:** 3D Manipulation, Multi-View, Transformer

## 1 Introduction

A fundamental goal of robot learning is to build systems that can solve various manipulation tasks in unconstrained 3D settings. A popular class of learning methods directly processes image(s) viewed from single or multiple cameras. These view-based methods have achieved impressive success on a variety of pick-and-place and object rearrangement tasks [1, 2, 3, 4]. However, their success on tasks that require 3D reasoning has been limited. As shown by James et al. [5] and Shridhar et al. [6], view-based methods struggle at 3D manipulation tasks on RLBench [7] with less than 2% success.

To address this, methods have been proposed that reason with explicit 3D representations of the scene. C2F-ARM [5] represents the scene with multi-resolution voxels and achieves strong performance on difficult RLBench tasks. PerAct [6] improves upon C2F-ARM in behavior cloning by using perceiver transformer [8] to process voxels. However, creating and reasoning over voxels comes at a higher computing cost compared to reasoning over images, since the number of voxels scales cubically with the resolution as opposed to squarely for image pixels. This makes voxel-based methods less scalable compared to their view-based counterparts. In fact, training PerAct on 18 RLBench tasks takes 16 days using 8 V100 GPUs (3072 GPU hours). This hinders fast development and prototyping. Moreover, such computing requirements become even more prohibitive when scaling to larger datasets with more tasks and diversity.

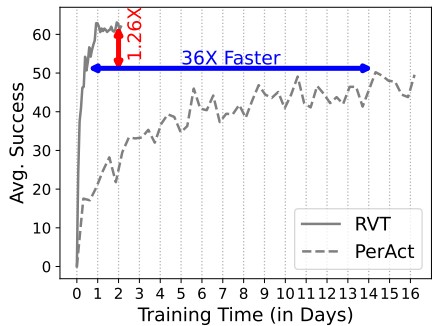

Figure 1: RVT scales and performs better than PerAct on RLBench, achieving on-par performance in 36X less time (same hardware), and 1.26X peak performance.

7th Conference on Robot Learning (CoRL 2023), Atlanta, USA.

Hence, a key question is – can we build a manipulation network that not only performs well but also inherits the scalability of view-based methods? To this end, we propose RVT (**R**obotic **V**iew **T**ransformer) that significantly outperforms the SOTA voxel-based method both in terms of success rate and training time, as shown in Fig. 1. With the same hardware, RVT achieves the peak performance of PerAct in **36X less time**, decreasing the training time from 14 days to just 10 hours. Apart from being much faster to train, RVT also achieves a **26% higher success rate** than PerAct, averaged over 18 tasks (249 task variations) on RLBench. RVT outperforms PerAct on **88.9% of tasks** on RLBench while achieving **2.3X the inference speed** (11.6 vs 4.9 fps). Further, we find that RVT also works well in the real world, where with only 51 demonstrations, a single RVT model can learn to perform a variety of manipulation tasks (5 tasks, 13 variations) like opening a drawer, placing objects on a shelf, pressing hand sanitizer, and stacking objects (see Fig. 4).

At its core, RVT is a view-based method that leverages the transformer architecture. It jointly attends over multiple views of the scene and aggregates information across the views. It then produces view-wise heatmaps and features that are used to predict robot end-effector pose. We extensively explore the design of the multi-view architecture and report several useful findings. For example, we observe a better performance when enforcing the transformer to first attend over patches within the same image before concatenating the patches for joint attention.

Another key innovation is that, unlike prior view-based methods, we decouple the camera images from the images fed to the transformer, by re-rendering the images from *virtual views*. This allows us to control the rendering process and leads to several benefits. For example, we can re-render from viewpoints that are useful for the task (e.g., directly above the table) while not being restricted by real-world physical constraints. Also, since the multi-view input to RVT is obtained via re-rendering, we can use RVT even with a single sensor camera – as done in our real-world experiments.

To summarize, our contributions are threefold: first, we propose RVT, a multi-view transformer for 3D object manipulation that is accurate and scalable; second, we investigate various design choices for the multi-view transformer that lead to better object manipulation performance; and finally, we present an empirical study for multi-task object manipulation in simulation and the real world.

## 2  Related Work

**Vision-based Object Manipulation.** The learning of robotic control policy has been traditionally studied with low-dimensional state observations [9, 10, 11, 12, 13]. Recently, vision-based policies [14, 15, 16, 17, 18, 19, 20, 21] have gained increasing attention since the high-dimensional visual sensory input provides more generalizable observation representation across tasks and is more accessible in real-world perception systems. Various forms of visual input have been explored. Prior work has directly encoded the RGB images into a low-dimensional latent space and relied on model-based [22, 23] or model-free [24, 25] reinforcement learning (RL) to train policies to operate in this space. More recently, RT-1 [26] infers the robot's actions from a history of images by leveraging transformer architectures [27]. Our proposed RVT also uses a transformer to predict actions, however, unlike RT-1, we additionally leverage depth to construct a multi-view scene representation. The use of depth input has also been extensively studied. Methods like CLIPort [3] and IFOR [1] directly process the RGB-D images for object manipulation, and hence are limited to simple pick-and-place tasks in 2D top-down settings. To overcome this issue, explicit 3D representations such as point clouds have been utilized. C2F-ARM [5] and PerAct [6] voxelize the point clouds and use a 3D convolutional network as the backbone for control inference. However, high-precision tasks typically require high resolution of voxelization, resulting in high memory consumption and slow training. Our approach falls into this category but addresses the scalability issue by transforming the point cloud into a set of RGB-D images from multiple views. We show that this significantly improves memory footprint and training efficiency, and leads to higher performance when compared to directly working with RGB(-D) or point cloud input (see Table. 1). Another relevant work is MIRA [28], which also uses novel view images to represent the 3D scene for action inference. MIRA achieves this by implicitly constructing a neural radiance field representation (NeRF) of the scene

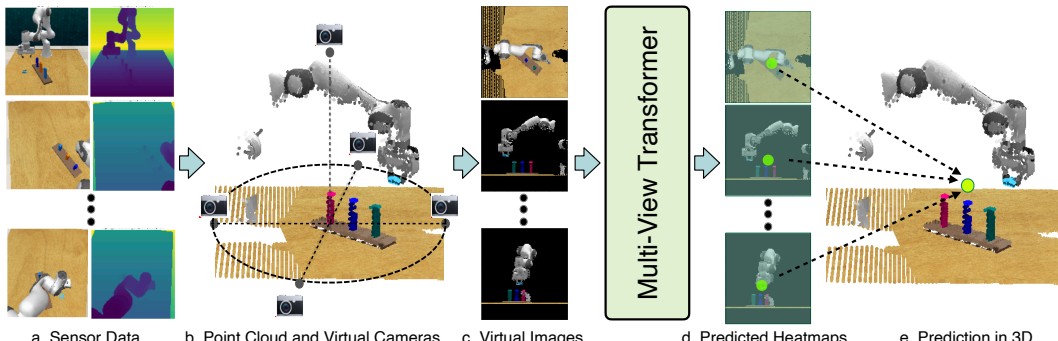

Figure 2: **Overview of RVT.** Given RGB-D from sensor(s), we first construct a point cloud of the scene. The point cloud is then used to produce virtual images around the robot workspace. The virtual images are fed to a multi-view transformer model to predict view-specific features, which are then combined to predict action in 3D.

from a set of RGB images and then generating novel view images from the optimized NeRF model. However, the requirement of optimizing a scene NeRF model slows down the inference speed at test time and relies on RGB images from a dense set of views as input. In contrast, our approach can achieve significantly faster inference speed and can work with even a single-view RGB image.

**Multi-Task Learning in Robotics.** Learning a single model for many different tasks has been of particular interest to the robotics community recently. A large volume of work achieves the multi-task generalization by using a generalizable task or action representation such as object point cloud [18, 19], semantic segmentation and optical flow [1], and object-centric representation [29, 30]. However, the limited expressiveness of such representations constrains them to only generalize within a task category. Task parameterization [31, 32] and discrete task-dependent output layer [33, 34] approaches are investigated with reinforcement learning to learn policies for tasks in different categories. With the recent breakthrough in large language models, multi-task robot learning has been approached by using natural language to specify a broad range of tasks and learning the policy from large pre-collected datasets [35, 26, 36, 2, 37, 38, 39, 40, 41]. We are inspired by this success but propose to learn language-conditioned multi-task policies with a small demonstration dataset.

**Transformers for Object Manipulation.** The success of transformers in vision and NLP has led its way into robot learning [42, 43, 44, 17, 45, 46]. Especially in object manipulation, transformer-based models with an attention mechanism can be utilized to extract features from sensory inputs to improve policy learning [47, 48, 49, 50, 51]. Unlike most prior work, we do not use large datasets for training. RVT efficiently learns from a small set of demonstrations, handle multiple views as visual inputs, and fuses information from language goals to tackle multiple manipulation tasks.

**Multi-View Networks in Computer Vision.** Multi-view representations have been explored in various vision problems. For point cloud recognition, SimpleView [52] showed how a simple view-based method outperforms sophisticated point-based methods. Follow-up works like MVTN [53] and Voint cloud [54] further improved upon SimpleView. Multi-view representations have also been used for other problems like 3D visual grounding [55], view synthesis [56] and depth prediction [57]. Unlike them, we focus on the problem of predicting robot actions for object manipulation.

## 3 Method

Our goal is to learn a single model that can complete a wide range of manipulation tasks. The input consists of (1) a language description of the task, (2) the current visual state (from RGB-D camera(s)), and (3) the current gripper state (open or closed). The model should predict an action, specified by a target end-effector pose and gripper state at the next *key-frame*. The key-frames represent important or bottleneck steps of the gripper during the task execution [58], such as a pre-pick, grasp, or place pose. Given a target end effector pose, we assume a low-level motion planner

and controller that can move the end effector to the target pose. To train the model, we assume a dataset $\mathcal{D} = \{D_1, D_2, \cdots, D_n\}$ of $n$ expert demonstrations covering various tasks is given. Each demonstration $D_i = (\{o^i_{1...m_i}\}, \{a^i_{1...m_i}\}, l_i)$ is a successful roll-out of length $m_i$, where $l_i$ is the language description of the task, $\{o^i_1, o^i_2, ..., o^i_{m_i}\}$ is a sequence of the observations from RGB-D camera(s) with gripper state, and $\{a^i_1, a^i_2, ..., a^i_{m_i}\}$ is the sequence of corresponding robot actions. This demonstration dataset can be used to train models with behavior cloning.

Our proposed method (RVT) is a transformer model [27] that processes images re-rendered around the robot workspace, produces an output for each view, and then back-projects into 3D to predict gripper pose actions, as shown in Fig. 2.

**Rendering.** The first step is the re-rendering of camera input. Given the RGB-D image(s) captured by one or multiple sensor cameras, we first reconstruct a point cloud of the scene. The point cloud is then re-rendered from a set of virtual viewpoints anchored in the space centered at the robot's base (see Fig. 2 and Fig. 3). Specifically, for each view, we render three image maps with a total of 7 channels: (1) RGB (3 channels), (2) depth (1 channel), and (3) $(x, y, z)$ coordinates of the points in the world frame (3 channels). The $(x, y, z)$ coordinates help establish the correspondence of pixels across views, i.e., if pixels from different views share the same $(x, y, z)$, they correspond to the same point in 3D. We use PyTorch3D [59] for rendering. We empirically verify various design choices in our rendering pipeline (see Tab. 2 (left)).

The re-rendering process decouples the input images to the ones fed to the transformer. This offers several benefits such as: the ability to re-render at arbitrary and useful locations (e.g., directly above the table) while not being constrained by real-world camera placements; multi-view reasoning even with a single sensor camera; allowing the use of orthographic images instead of generally provided perspective ones; facilitating 3D point-cloud augmentations and enabling additional channels like point correspondence which are not natively presented in the sensor images. We empirically find that these contribute to achieving high performance with view-based networks (see Sec. 4.1).

**Joint Transformer.** The re-rendered images, the language description of the task, and the gripper state (open or close) are processed by a joint transformer model (see Fig. A1 in the appendix). For language, we use pretrained CLIP [60] embeddings (ResNet-50 variant), which provide one token for each word. For the virtual images, we break each of them into $20 \times 20$ patches and pass through a multi-layer perceptron (MLP) to produce image tokens, similar to ViT [61]. For the gripper state, similar to PerAct [6], we pass it through an MLP and concatenate it to the image tokens. We also add positional embeddings to all the image and language tokens to preserve the positional information.

Overall, RVT has eight self-attention layers. In the first four layers, an image token is only allowed to attend to other tokens from the same image. This biases the network to process individual images first before sharing information across images. We concatenate all the image tokens along with the language tokens afterward. In the last four layers, we allow the attention layers to propagate and accumulate information across different images and text. Finally, the image tokens are rearranged back to the original spatial configuration, resulting in the feature channels of each image.

**Action Prediction.** The model outputs an 8D action, including the 6-DoF target end effector pose (3-DoF for translation and 3-DoF for rotation), 1-DoF gripper state (open or close), and a binary indicator for whether to allow collision for the low-level motion planner (see [6] for details). For translation, we first predict a heatmap for each view from the per-image features from the joint transformer (as shown in Fig. A1 in the appendix). The heatmaps across different views are then back-projected to predict scores for a discretized set of 3D points that densely cover the robot workspace (see Sec. A.3 in the appendix). Finally, the end effector translation is determined by the 3D point with the highest score. Note that this multi-view heatmap representation for translation prediction extends prior approaches in the 2D top-down view setting [4]. Hence, RVT inherits the benefit of superior sample efficiency by representing the visual input and action in the same spatial structure [4].

For end effector rotation, we follow PerAct to use the Euler angles representation, where each angle is discretized into bins of $5°$ resolution. The gripper state and the motion planner collision indicator are represented as binary variables. To predict the rotations, gripper state, and collision indicator,

| Models | Avg. Success ↑ | Avg. Rank ↓ | Train time (in days) ↓ | Inf. Speed (in fps) ↑ | Close Jar | Drag Stick | Insert Peg | Meat off Grill | Open Drawer | Place Cups | Place Wine |
|---|---|---|---|---|---|---|---|---|---|---|---|
| Image-BC (CNN) [2, 6] | 1.3 | 3.7 | - | - | 0 | 0 | 0 | 0 | 4 | 0 | 0 |
| Image-BC (ViT) [2, 6] | 1.3 | 3.8 | - | - | 0 | 0 | 0 | 0 | 0 | 0 | 0 |
| C2F-ARM-BC [5, 6] | 20.1 | 3.1 | - | - | 24 | 24 | 4 | 20 | 20 | 0 | 8 |
| PerAct [6] | 49.4 | 1.9 | 16.0 | 4.9 | **55.2** ± 4.7 | 89.6 ± 4.1 | 5.6 ± 4.1 | 70.4 ± 2.0 | **88.0** ± 5.7 | 2.4 ± 3.2 | 44.8 ± 7.8 |
| RVT (ours) | **62.9** | **1.1** | **1.0** | **11.6** | 52.0 ± 2.5 | **99.2** ± 1.6 | **11.2** ± 3.0 | **88.0** ± 2.5 | 71.2 ± 6.9 | **4.0** ± 2.5 | **91.0** ± 5.2 |

| Models | Push Buttons | Put in Cupboard | Put in Drawer | Put in Safe | Screw Bulb | Slide Block | Sort Shape | Stack Blocks | Stack Cups | Sweep to Dustpan | Turn Tap |
|---|---|---|---|---|---|---|---|---|---|---|---|
| Image-BC (CNN) [2, 6] | 0 | 0 | 8 | 4 | 0 | 0 | 0 | 0 | 0 | 0 | 8 |
| Image-BC (ViT) [2, 6] | 0 | 0 | 0 | 0 | 0 | 0 | 0 | 0 | 0 | 0 | 16 |
| C2F-ARM-BC [5, 6] | 72 | 0 | 4 | 12 | 8 | 16 | 8 | 0 | 0 | 0 | 68 |
| PerAct [6] | 92.8 ± 3.0 | 28.0 ± 4.4 | 51.2 ± 4.7 | 84.0 ± 3.6 | 17.6 ± 2.0 | 74.0 ± 13.0 | 16.8 ± 4.7 | 26.4 ± 3.2 | 2.4 ± 2.0 | 52.0 ± 0.0 | 88.0 ± 4.4 |
| RVT (ours) | **100.0** ± 0.0 | **49.6** ± 3.2 | **88.0** ± 5.7 | **91.2** ± 3.0 | **48.0** ± 5.7 | **81.6** ± 5.4 | **36.0** ± 2.5 | **28.8** ± 3.9 | **26.4** ± 8.2 | **72.0** ± 0.0 | **93.6** ± 4.1 |

Table 1: **Multi-Task Performance on RLBench.** RVT outperforms state-of-the-art methods while being faster to train and execute. RVT has the best success rate and rank when averaged across all tasks. Performance for Image-BC (CNN), Image-BC (ViT) and C2F-ARM-BC are as reported by Shridhar et al. in [6]. We re-evaluate PerAct using the released final model and estimate mean and variance. RVT is 2.3X faster on execution speed than PerAct and outpeforms it on 16/18 tasks. The training time and inference speed of PerAct and RVT are measured on the same GPU model.

we use global features ($\mathcal{G}$). The global features are a concatenation of (1) the sum of image features along the spatial dimensions, weighted by the predicted translation heatmap; and (2) the max-pooled image features along the spatial dimension. Specifically, let $f_i$ be the image feature and $h_i$ be the predicted translation heatmap for the $i$th image. Then the global feature $\mathcal{G}$ is given by $\mathcal{G} = [\phi(f_1 \odot h_1); \cdots ; \phi(f_K \odot h_K); \psi(f_1); \cdots ; \psi(f_K)]$, where $K$ is the number of images, $\odot$ denotes element-wise multiplication, and $\phi$ and $\psi$ denote the sum and max-pooling over the height and width dimensions. The weighted sum operation provides higher weights to image locations near the predicted end effector position.

**Loss Function.** We train RVT using a mixture of losses. For heatmaps, we use the cross-entropy loss for each image. The ground truth is obtained by a truncated Gaussian distribution around the 2D projection of the ground-truth 3D location. For rotation, we use the cross-entropy loss for each of the Euler angles. We use binary classification loss for the gripper state and collision indicator.

## 4 Experiments

### 4.1 Simulation Experiments

**Simulation Setup.** We follow the simulation setup in PerAct [6], where CoppeliaSim [62] is applied to simulate various RLBench [7] tasks. A Franka Panda robot with a parallel gripper is controlled to complete the tasks. We test on the same 18 tasks as PerAct, including picking and placing, tool use, drawer opening, and high-accuracy peg insertions (see the appendix for a detailed specification of each task). Each task includes several variations specified by the associated language description. Such a wide range of tasks and intra-task variations requires the model to not just specialize in one specific skill but rather learn different skill categories. The visual observations are captured from four noiseless RGB-D cameras positioned at the front, left shoulder, right shoulder, and wrist with a resolution of $128 \times 128$. To achieve the target gripper pose, we generate joint space actions by using the same sampling-based motion planner [63, 64] as in [5, 6].

**Baselines.** We compare against the following three baselines: (1) *Image-BC* [2] is an image-to-action behavior cloning agent that predicts action based on the image observations from the sensor camera views. We compare with two variants with CNN and ViT vision encoders respectively. (2) *C2F-ARM-BC* [5] is a behavior cloning agent that converts the RGB-D images into multi-resolution voxels and predicts the next key-frame action using a coarse-to-fine scheme. (3) *PerAct* [6] is the state-of-the-art multi-task behavior cloning agent that encodes the RGB-D images into voxel grid patches and predicts discretized next key-frame action using the perceiver [8] transformer.

**Training and Evaluation Details.** Just like the baselines, we use the RLBench training dataset with 100 expert demonstrations per task (1800 demonstrations over all tasks). Similar to PerAct, we apply

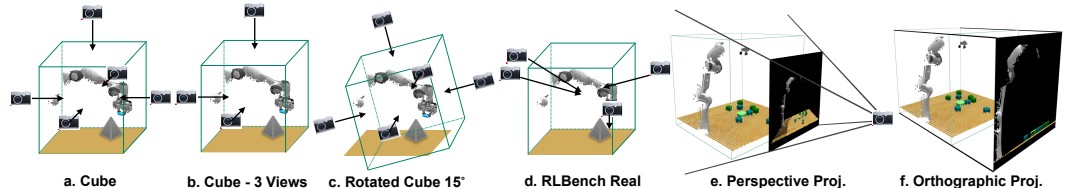

Figure 3: We evaluate RVT with various camera locations for re-rendering (a-d) and find that locations in (a) perform best. We also test various projection options (e-f) for rendering images and find that RVT works better with orthographic images.

translation and rotation data augmentations. For translation, we randomly perturb the point clouds in the range $[\pm 0.125m, \pm 0.125m, \pm 0.125m]$. For rotation, we randomly rotate the point cloud around the $z$-axis (vertical) in the range of $\pm 45°$. We train RVT for 100k steps, using the LAMB [65] optimizer as PerAct. We use a batch size of 24 and an initial learning rate of $2.4 \times 10^{-4}$. We use cosine learning rate decay with warm-start for 2K steps.

For Image-BC and C2F-ARM-BC, we adopt the evaluation results from [6] since their trained models have not been released. These results overestimate the performance of Image-BC and C2F-ARM-BC, as they select the best model for each of the 18 tasks independently based on the performance on validation sets. Hence, the reported performance does not reflect a single multi-task model. Nevertheless, these baselines still underperform both PerAct and RVT (see Tab. 1). For PerAct, we evaluate the final model released by Shridhar et al. [6]. We test our models (including the models in the ablation study, Tab. 2 (left)) and PerAct on the same 25 variations for each task. Due to the randomness of the sampling-based motion planner, we run each model five times on the same 25 variations for each task and report the average success rate and standard deviation in Tab. 1.

To fairly compare the training efficiency against PerAct, we train both PerAct and our model with the same GPU type (NVIDIA Tesla V100) and number of GPUs (8), as reported by Shridhar et al. [6]. We report the total training time for both models in Tab. 1 ("Training time"). We also evaluate the inference speed of PerAct and RVT models by running the prediction inferences for the same input data on the same GPU (NVIDIA RTX 3090).

**Multi-Task Performance.** Tab. 1 compares the performance between RVT and the baselines. We find that PerAct and RVT perform significantly better than the rest. Overall, RVT outperforms all baselines with the best rank and success rate when averaged across all tasks. It outperforms prior state-of-the-art methods, C2F-ARM, by 42 percentage points (213% relative improvement); and PerAct by 13 percentage points (26% relative improvement). RVT outperforms PerAct on 88.9% (16/18) of the tasks. More remarkably, RVT trains 36X faster than PerAct for achieving the same performance (see Fig. 1). We also observe that at inference time, RVT is 2.3X faster than PerAct. These results demonstrate that RVT is both more accurate and scalable when compared to existing state-of-the-art voxel-based methods.

**Ablation Study.** We conduct ablation experiments to analyze different design choices of RVT: (a) the resolution of the rendered images ("Im. Res." column in Tab. 2 (left)); (b) whether to include the correspondence information across rendered images ("View Corr."); (c) whether to include the depth channel ("Dep. Ch."); (d) whether to separately process the tokens of each image before jointly processing all tokens ("Sep. Proc."); (e) the projection type for rendering—perspective or orthographic ("Proj. Type"); (f) whether to use rotation augmentation ("Rot. Aug."); (g) the number of views and camera locations for re-rendering ("# of View" and "Cam. Loc."); and (h) the benefit of using re-rendered images versus using real sensor camera images ("Real" for "Cam. Loc.").

Tab. 2 (left) summarizes the ablation experiment results. The same table along with the mean and standard dev. for each task can be found in the appendix Tab. A2. Below we discuss the findings:

(a) As expected, virtual images rendered at higher resolution help as RVT with virtual image resolution 220 outperforms the one with 100.

| Im. Res. | View Corr. | Dep. Ch. | Sep. Proc. | Proj. Type | Rot. Aug. | Cam Loc. | # of View | Avg. Succ. |
|---|---|---|---|---|---|---|---|---|
| 220 | ✓ | ✓ | ✓ | Orth. | ✓ | Cube | 5 | **62.9** |
| 100 | ✓ | ✓ | ✓ | Orth. | ✓ | Cube | 5 | 51.1 |
| 220 | ✗ | ✓ | ✓ | Orth. | ✓ | Cube | 5 | 59.7 |
| 220 | ✓ | ✗ | ✓ | Orth. | ✓ | Cube | 5 | 60.3 |
| 220 | ✓ | ✓ | ✗ | Orth. | ✓ | Cube | 5 | 58.4 |
| 220 | ✓ | ✓ | ✓ | Pers. | ✓ | Cube | 5 | 40.2 |
| 220 | ✓ | ✓ | ✓ | Orth. | ✗ | Cube | 5 | 60.4 |
| 220 | ✓ | ✓ | ✓ | Orth. | ✓ | Cube | 3 | 60.2 |
| 220 | ✓ | ✓ | ✓ | Orth. | ✓ | Front | 1 | 35.8 |
| 220 | ✓ | ✓ | ✓ | Orth. | ✓ | Rot. 15 | 5 | 59.9 |
| 220 | ✓ | ✓ | ✓ | Pers. | ✗ | Real | 4 | 10.4 |
| 220 | ✓ | ✓ | ✓ | Orth. | ✗ | Real | 4 | 22.9 |

| Task | # of vari. | # of train | # of test | PerAct (+ mark.) | RVT (+ mark.) | PerAct (- mark.) | RVT (- mark.) |
|---|---|---|---|---|---|---|---|
| Stack blocks | 3 | 14 | 10 | 50% | 100% | 50% | 100% |
| Press sanitizer | 1 | 7 | 10 | 40% | 80% | 40% | 80% |
| Put marker in mug/bowl | 4 | 12 | 10 | 0% | 0% | – | – |
| Put object in drawer | 3 | 10 | 10 | 20% | 50% | 50% | 100% |
| Put object in shelf | 2 | 8 | 10 | 30% | 50% | 30% | 50% |
| All tasks | 13 | 51 | 50 | 28% | 56% | 42.5% | 82.5% |

Table 2: **Left:** Ablations on RLBench. A larger res., adding view correspondence, adding depth channel, separating initial attention layers, orthographic projection, using rotation aug., and re-rendered views around cube improve performance. **Right:** Success rate of RVT and Peract in the real-world. A single RVT model can perform well on most tasks with only a few demonstrations.

(b) Adding correspondence information for points across different views helps (see Sec. 3). This is likely because the network need not learn to solve the correspondence problem and can predict more consistent heatmaps across views. Note that the view correspondence channel is not present in sensor images but is rendered along with RGB(D) images in RVT.

(c) Adding the depth channel along with RGB channels helps, likely because it aids 3D reasoning.

(d) Independently processing the tokens from a single image, before merging all the image tokens, helps. It is likely because this design expects the network to extract meaningful features for each image before reasoning over them jointly.

(e) Rendering images with orthographic projection performs better than rendering with perspective projection, for both the cube and real camera locations. We hypothesize that it is because orthographic projection preserves the shape and size of an object regardless of its distance from the camera (see Fig. 3 (e-f)). It also highlights the advantage of re-rendering, as real sensors generally render with perspective projections.

(f) As expected, using 3D rotation augmentation in the point cloud before rendering helps. To take advantage of 3D augmentations, the re-rendering process is necessary.

(g) The model with 5 views around a cube (Fig. 3 (a)) performs the best followed by the one with 3 views (front, top, left) around a cube (Fig. 3 (b)). The single view model, where we predict the third coordinate as an offset like TransporterNet [4], performs substantially worse, calling for the need for multiple views for 3D manipulation. It also highlights the advantage of re-rendering as with re-rendering we can leverage multiple views even with a single sensor camera. We also empirically find that rotating the location of the cameras by $15°$ (see Fig. 3) with respect to the table (and robot) decreases performance. This could be likely because views aligned with the table and robot might be easier to reason with (e.g., overhead top view, aligned front view).

(h) RVT performs better with re-rendered images as compared to using sensor camera images (Tab. 2 (left), second last row). The sensor camera images are rendered with perspective projection (physical rendering process) and are not straightforward to apply 3D augmentations (e.g., rotation) without re-rendering. Also, the location of sensor cameras may be sub-optimal for 3D reasoning, e.g., the views are not axially aligned with the table or robot (see Fig. 3 (d)). All these factors contribute to RVT performing better with re-rendered images than with sensor camera images.

Notably, one might consider rearranging the sensor cameras to match the re-rendering views in order to bypass re-rendering (see appendix A.2). However, this will void the gains from using orthographic projections, 3D augmentation, and adding correspondences (see appendix A.3). This also strictly requires a multi-camera setup (Fig. 3 (a)), which is more costly and less portable in the real world than using one sensor camera. Finally, we have briefly explored view selection and found an option that works well across tasks. Further optimization of views, including the sensor and re-rendered ones, is an interesting future direction.

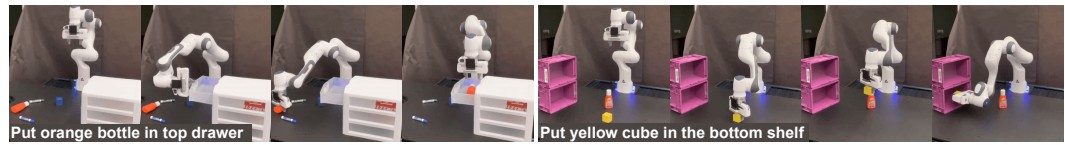

Figure 4: **Examples of RVT in the real world.** A single RVT model can perform multiple tasks (5 tasks, 13 variations) in the real world with just ∼10 demonstrations per task.

## 4.2 Real-World

**Real World Setup.** We experiment on a table-top setup using a statically mounted Franka Panda arm. The scene is perceived via an Azure Kinect (RGB-D) camera statically mounted in a third-person view. We calibrate the robot-camera extrinsics and transform the perceived point clouds to the robot base frame before passing into RVT. Given a target gripper pose from RVT, we use FrankaPy [66] to move the robot to the target with trajectory generation and feedback control.

**Tasks.** We adopt a total of 5 tasks similar to the ones in PerAct [6] (see Tab. 2 (right)): *stack blocks*, *press sanitizer*, *put marker in mug/bowl*, *put object in drawer*, *put object in shelf*. Each task can be instantiated with different *variations* defined by the language description. For example, for *stack blocks*, some variations could be "put yellow block on blue block" and "put blue block on red block". Given a task and variation, we sample a scene by placing the task-related objects and a set of distractor objects on the table in a random configuration.

**Data Collection.** We first collect a dataset for training RVT through human demonstration. Given a sampled task and scene configuration, we ask the human demonstrator to specify a sequence of gripper target poses by kinesthetically moving the robot arm around. Once we have the target pose sequence, we reset the robot to the start pose, and then control it to sequentially move to each target pose following the specified order. We simultaneously record the RGB-D stream from the camera during the robot's motion to the targets. This provides us with a dataset of RGB-D frames paired with target pose annotations. In total, we collected 51 demonstration sequences over all 5 tasks.

**Results.** We train on real-world data for 10K steps, with the same optimizer, batch size, and learning rate schedule as the simulation data. We report the results in Tab. 2 (right). We find RVT outperforms prior method PerAct. It achieves high success rates for the *stack block* task (100%) and the *press sanitizer* task (80%). Even on longer horizon tasks such as putting objects in drawers and shelves (e.g., the robot has to first open the drawer/shelf and then pick up the object), our model achieves 50% success rates (see Fig. 4). We found RVT struggled with marker-related tasks, which is likely due to sparse and noisily sensed point clouds. We further divide the results into two sets: "+ markers" (full set) and "- markers". Our model overall achieves an 82.5% success rate on non-marker tasks. The marker issue can potentially be addressed by attaching the camera to the gripper to capture point clouds at higher quality. Another possibility is to use zoom-in views similar to C2F-ARM [5].

## 5 Conclusions and Limitations

We proposed RVT, a multi-view transformer model for 3D object manipulation. We found that RVT outperforms prior state-of-the-art models like PerAct and C2F-ARM on a variety of 3D manipulation tasks, while being more scalable and faster. We also found that RVT can work on real-world manipulation tasks with only a few demonstrations.

Although we found RVT achieves state-of-the-art results on RLBench (62.9% success rate), there is scope for improvement. Following, we identify some limitations that present exciting directions for future research. We briefly explore various options for virtual view and found the orthogonal views work well across tasks, but it would be exciting if the virtual views can be optimized further or learned from data. Further, when compared to prior view-based methods, RVT (as well as explicit voxel-based methods like PerAct and C2F-ARM), requires the calibration of extrinsics from the camera to the robot base. It would be exciting to explore extensions that remove this constraint.

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

# A   Appendix

## A.1   RLBench Tasks

We provide a brief summary of the RLBench tasks in Tab. A1. There are 18 tasks with 249 variations. For more detailed description of each task, please refer to PerAct [6], Appendix A.

| Task | Language Template | # of Variations |
|---|---|---|
| open drawer | "open the __ drawer" | 3 |
| slide block | "slide the __ block to target" | 4 |
| sweep to dustpan | "sweep dirt to the __ dustpan" | 2 |
| meat off grill | "take the __ off the grill" | 2 |
| turn tap | "turn __ tap" | 2 |
| put in drawer | "put the item in the __ drawer" | 3 |
| close jar | "close the __ jar" | 20 |
| drag stick | "use the stick to drag the cube onto the __ target" | 20 |
| stack blocks | "stack __ __ blocks" | 60 |
| screw bulb | "screw in the __ light bulb" | 20 |
| put in safe | "put the money away in the safe on the __ shelf" | 3 |
| place wine | "stack the wine bottle to the __ of the rack" | 3 |
| put in cupboard | "put the __ in the cupboard" | 9 |
| sort shape | "put the __ in the shape sorter" | 5 |
| push buttons | "push the __ button, [then the __ button]" | 50 |
| insert peg | "put the __ peg in the spoke" | 20 |
| stack cups | "stack the other cups on top of the __ cup" | 20 |
| place cups | "place __ cups on the cup holder" | 3 |

Table A1: **Tasks in RLBench** We evaluate on 18 RLBench tasks which are same as those used in PerAct [6]. For more details, check see PerAct [6], Appendix A. For videos, visit https://robotic-view-transformer.github.io/

## A.2   Additional Experiments

**Experiments on dataset with input cameras in orthogonal configuration.** We created a new image dataset for RLBench with input cameras arranged in orthogonal configuration like in Fig. 3a. Note that this dataset is not provided in PerAct. With this new image dataset, we did two experiments, one directly using input camera images as input and one with our pipeline (re-rendering with orthographic projection, 3D augmentation and correspondences). We find that results on this new dataset are consistent with the results on the dataset provided by PerAct, where our pipeline works better (60.0% vs 27.2%) likely because it allows for orthographic projection, 3D augmentation, and point correspondence (Table 2 Left).

**How does the quality of input point cloud affect performance of RVT?** To investigate how the quality of the point cloud affects performance in RVT, we do experiments in simulation. Specifically, we add Gaussian noise with varying standard deviation (2.5mm, 5mm, 1cm, 2cm, and 4cm) to the original point cloud. We add this noise both during training and evaluation to simulate sensor noise in both phases. The success rate is 62.9 for no noise, 62.0 for 2.5mm, 61.6 for 5mm, 56.4 for 1cm, 58.7 for 2cm and 51.7 for 4cm standard deviation noise. We find that RVT is robust to 2cm standard deviation noise in the point cloud and its performance degrades gracefully with more noise. For reference, the depth measurements in Intel RealSense D400 camera has an error of 2.5mm to 5mm for an object at 1m from the camera (source: https://www.intel.com/content/www/us/en/support/articles/000026260/emerging-technologies/intel-realsense-technology.html)

### A.3   Additional Explanation

**How are the heatmaps of multiple virtual views back-projected to 3D?** To calculate the heatmap value at a 3D location, we map 3D points to 2D pixels in the virtual views. We consider not just the points in the point cloud, but all points in the 3D scene bounds that are distributed at a resolution of $(h \times h \times h)$, where $(h \times h)$ is the resolution of the virtual image. For each point, the heatmap value from multiple views are averaged.

**Why only using input images (without re-renderning) voids the gains from using orthographic projections, 3D augmentation, and adding correspondences?** As we see in Table 2 Left, orthographic projections, 3D augmentation and adding xyz image (correspondence) improve performance. However, these could only be added after re-rendering, because: First, real-world cameras generally provide only perspective projection and not orthographic projection. To obtain orthographic projection, re-rendering is needed. Second, effects of 3D augmentation like rotation of the object cannot be trivially simulated in the image without re-rendering. We first create a 3D representation, apply the augmentation and re-render. Finally, adding xyz image between points in images first requires explicitly building the 3D point cloud from images, and rendering the xyz image.

**In the real robot experiments, only one camera view is used. How carefully does this view have to be selected so that the method performs well? Would the performance of the method improve if more cameras are placed in the workspace?** We used a standard third person view that is in front of the robot. We ensured that the workspace is visible in the camera but no particular effort was put in adjusting the view to make the method perform well. Potentially having more cameras could improve the method.

**PerAct extracts a set of keyframe actions from the demo by capturing bottleneck end-effector poses in the action sequence that has (1) near-zero joint velocities or (2) an unchanged gripper open state. It seems from Sec 4.2 (Data Collection) that the keyframe actions are specified in a different way in RVT.** We follow the same pipeline as PerAct to extract keyframes from demonstrations. Sec 4.2 (Data Collection) describes our scheme for collecting target poses for real world demonstrations. The difference is that PerAct uses a VR controller to specify target poses while we do so by kinesthetically moving the robot. Once a real-world demonstration is collected, the process of extracting keyframes is the same.

## A.4  RVT Overview

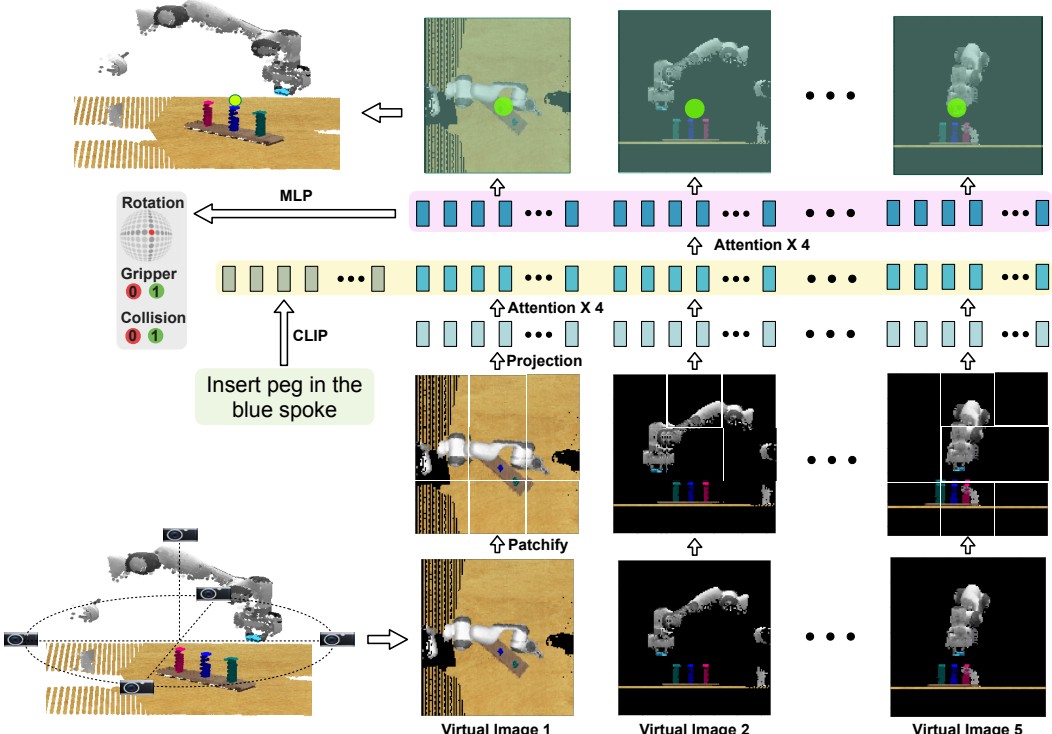

Figure A1: **Overview of the transformer used in RVT.** The input to the transformer is a language description of the task and virtual images of the scene point cloud. The text is converted into token embeddings using the pretrained CLIP [60] model, while the virtual images are converted into token embeddings via patchify and projection operations. For each virtual image, tokens belonging to the same image are processed via four attention layers. Finally, the processed image tokens as well as the language tokens are jointly processed using four attention layers. The 3D action is inferred using the resulting image tokens.

## A.5 Ablations

We report the ablations mentioned in Tab. 2, along with the mean and standard deviations for each task in Tab. A2.

| Im. Res. | View Corr. | Dep. Ch. | Bi-Lev. | Proj. Type | Rot. Aug. | Cam Loc. | # of View | Avg. Succ. | Close Jar | Drag Stick | Insert Peg | Meat off Grill | Open Drawer | Place Cups |
|---|---|---|---|---|---|---|---|---|---|---|---|---|---|---|
| 220 | ✓ | ✓ | ✓ | Orth. | ✓ | Cube | 5 | **62.9** | 52 ± 2.5 | 99.2 ± 1.6 | 11.2 ± 3 | 88 ± 2.5 | 71.2 ± 6.9 | 4 ± 2.5 |
| 100 | ✓ | ✓ | ✓ | Orth. | ✓ | Cube | 5 | 51.1 | 60 ± 0 | 83 ± 1.7 | 4 ± 2.8 | 91 ± 3.3 | 67 ± 5.2 | 1 ± 1.7 |
| 220 | ✗ | ✓ | ✓ | Orth. | ✓ | Cube | 5 | 59.7 | 44 ± 0 | 100 ± 0 | 17 ± 4.4 | 90 ± 6 | 71 ± 9.1 | 7 ± 5.9 |
| 220 | ✓ | ✗ | ✓ | Orth. | ✓ | Cube | 5 | 60.3 | 37 ± 3.3 | 96 ± 0 | 11 ± 3.3 | 97 ± 1.7 | 57 ± 8.2 | 3 ± 3.3 |
| 220 | ✓ | ✓ | ✗ | Orth. | ✓ | Cube | 5 | 58.4 | 32 ± 7.5 | 96 ± 0 | 11 ± 3.3 | 90 ± 2 | 68 ± 2.8 | 2 ± 2 |
| 220 | ✓ | ✓ | ✓ | Pers. | ✓ | Cube | 5 | 40.2 | 20 ± 2.5 | 90.4 ± 2 | 4 ± 0 | 84.8 ± 4.7 | 13.6 ± 4.8 | 2.4 ± 2 |
| 220 | ✓ | ✓ | ✓ | Orth. | ✗ | Cube | 5 | 60.4 | 52 ± 0 | 92 ± 0 | 12.8 ± 1.6 | 97.6 ± 4.8 | 85.6 ± 5.4 | 0 ± 0 |
| 220 | ✓ | ✓ | ✓ | Orth. | ✓ | Cube | 3 | 60.2 | 44.8 ± 1.6 | 75.2 ± 4.7 | 15 ± 3.3 | 89.6 ± 4.1 | 68.8 ± 9.3 | 3.2 ± 1.6 |
| 220 | ✓ | ✓ | ✓ | Orth. | ✓ | Front | 1 | 35.8 | 36 ± 4.9 | 87 ± 1.7 | 2 ± 2 | 90 ± 6 | 58 ± 6.6 | 0 ± 0 |
| 220 | ✓ | ✓ | ✓ | Orth. | ✓ | Rot. 15 | 5 | 59.9 | 48.8 ± 1.6 | 99.2 ± 1.6 | 12 ± 4.4 | 80 ± 2.5 | 71.2 ± 9.3 | 0 ± 0 |
| 220 | ✓ | ✓ | ✓ | Pers. | ✗ | Real | 4 | 10.4 | 14.4 ± 6.5 | 14.4 ± 5.4 | 0 ± 0 | 0 ± 0 | 22.4 ± 5.4 | 0 ± 0 |
| 220 | ✓ | ✓ | ✓ | Ortho. | ✗ | Real | 4 | 22.9 | 43.2 ± 4.7 | 54.4 ± 3.2 | 0 ± 0 | 0 ± 0 | 15.2 ± 5.3 | 0.8 ± 1.6 |

| Im. Res. | View Corr. | Dep. Ch. | Bi-Lev. | Proj. Type | Rot. Aug. | Cam Loc. | # of View | Avg. Succ. | Place Wine | Push Buttons | Put in Cupboard | Put in Drawer | Put in Safe | Screw Bulb |
|---|---|---|---|---|---|---|---|---|---|---|---|---|---|---|
| 220 | ✓ | ✓ | ✓ | Orth. | ✓ | Cube | 5 | **62.9** | 91 ± 5.2 | 100 ± 0 | 49.6 ± 3.2 | 88 ± 5.7 | 91.2 ± 3 | 48 ± 5.7 |
| 100 | ✓ | ✓ | ✓ | Orth. | ✓ | Cube | 5 | 51.1 | 38 ± 8.7 | 100 ± 0 | 49 ± 4.4 | 86 ± 2 | 77 ± 1.7 | 22 ± 4.5 |
| 220 | ✗ | ✓ | ✓ | Orth. | ✓ | Cube | 5 | 59.7 | 96 ± 2.8 | 99 ± 1.7 | 48 ± 6.9 | 50 ± 6 | 79 ± 5.9 | 36 ± 0 |
| 220 | ✓ | ✗ | ✓ | Orth. | ✓ | Cube | 5 | 60.3 | 71 ± 1.7 | 99 ± 1.7 | 56 ± 0 | 92 ± 4.9 | 77 ± 3.3 | 39 ± 4.4 |
| 220 | ✓ | ✓ | ✗ | Orth. | ✓ | Cube | 5 | 58.4 | 65 ± 5.2 | 100 ± 0 | 54 ± 2 | 94 ± 4.5 | 78 ± 3.5 | 48 ± 6.3 |
| 220 | ✓ | ✓ | ✓ | Pers. | ✓ | Cube | 5 | 40.2 | 28 ± 5.7 | 91.2 ± 1.6 | 26.4 ± 2 | 64.8 ± 3 | 51.2 ± 3.9 | 20 ± 4.4 |
| 220 | ✓ | ✓ | ✓ | Orth. | ✗ | Cube | 5 | 60.4 | 84 ± 3.6 | 96 ± 2.5 | 40 ± 2.5 | 88 ± 7.2 | 90.4 ± 4.1 | 48 ± 8.4 |
| 220 | ✓ | ✓ | ✓ | Orth. | ✓ | Cube | 3 | 60.2 | 84.8 ± 8.9 | 97.6 ± 2 | 40.8 ± 4.7 | 94.4 ± 4.1 | 82.4 ± 7.8 | 43.2 ± 3.9 |
| 220 | ✓ | ✓ | ✓ | Orth. | ✓ | Front | 1 | 35.8 | 82 ± 4.5 | 46 ± 2 | 14 ± 4.5 | 29 ± 7.1 | 57 ± 5.9 | 6 ± 2 |
| 220 | ✓ | ✓ | ✓ | Orth. | ✓ | Rot. 15 | 5 | 59.9 | 74.4 ± 5.4 | 99.2 ± 1.6 | 46.4 ± 4.1 | 81.6 ± 2 | 80.8 ± 4.7 | 45.6 ± 4.8 |
| 220 | ✓ | ✓ | ✓ | Pers. | ✗ | Real | 4 | 10.4 | 11.2 ± 3.9 | 26.4 ± 4.1 | 0 ± 0 | 0 ± 0 | 0 ± 0 | 0 ± 0 |
| 220 | ✓ | ✓ | ✓ | Ortho. | ✗ | Real | 4 | 22.9 | 67.2 ± 5.9 | 76 ± 5.7 | 0 ± 0 | 0 ± 0 | 0 ± 0 | 0 ± 0 |

| Im. Res. | View Corr. | Dep. Ch. | Bi-Lev. | Proj. Type | Rot. Aug. | Cam Loc. | # of View | Avg. Succ. | Slide Block | Sort Shape | Stack Blocks | Stack Cups | Sweep to Dustpan | Turn Tap |
|---|---|---|---|---|---|---|---|---|---|---|---|---|---|---|
| 220 | ✓ | ✓ | ✓ | Orth. | ✓ | Cube | 5 | **62.9** | 81.6 ± 5.4 | 36 ± 2.5 | 28.8 ± 3.9 | 26.4 ± 8.2 | 72 ± 0 | 93.6 ± 4.1 |
| 100 | ✓ | ✓ | ✓ | Orth. | ✓ | Cube | 5 | 51.1 | 93 ± 3.3 | 18 ± 2 | 17 ± 5.2 | 1 ± 1.7 | 36 ± 0 | 76 ± 2.8 |
| 220 | ✗ | ✓ | ✓ | Orth. | ✓ | Cube | 5 | 59.7 | 83 ± 1.7 | 41 ± 4.4 | 26.7 ± 5 | 20 ± 4.9 | 72 ± 0 | 95 ± 4.4 |
| 220 | ✓ | ✗ | ✓ | Orth. | ✓ | Cube | 5 | 60.3 | 72 ± 4 | 37 ± 5.2 | 23 ± 3.3 | 33 ± 5.9 | 92 ± 0 | 95 ± 4.4 |
| 220 | ✓ | ✓ | ✗ | Orth. | ✓ | Cube | 5 | 58.4 | 66 ± 6 | 31 ± 6.6 | 25 ± 3.3 | 29 ± 5.2 | 72 ± 0 | 91 ± 3.3 |
| 220 | ✓ | ✓ | ✓ | Pers. | ✓ | Cube | 5 | 40.2 | 88 ± 4.4 | 19.2 ± 4.7 | 22.4 ± 9 | 1.6 ± 2 | 16 ± 0 | 80.8 ± 3 |
| 220 | ✓ | ✓ | ✓ | Orth. | ✗ | Cube | 5 | 60.4 | 72.8 ± 1.6 | 25.6 ± 2 | 18.4 ± 6 | 8.8 ± 5.3 | 84 ± 0 | 92 ± 2.5 |
| 220 | ✓ | ✓ | ✓ | Orth. | ✓ | Cube | 3 | 60.2 | 95.2 ± 1.6 | 37.6 ± 4.1 | 29.6 ± 3.2 | 8.8 ± 4.7 | 80 ± 0 | 92.8 ± 3 |
| 220 | ✓ | ✓ | ✓ | Orth. | ✓ | Front | 1 | 35.8 | 42 ± 2 | 2 ± 2 | 0 ± 0 | 0 ± 0 | 0 ± 0 | 93 ± 5.2 |
| 220 | ✓ | ✓ | ✓ | Orth. | ✓ | Rot. 15 | 5 | 59.9 | 83 ± 1.7 | 30.4 ± 5.4 | 46.4 ± 9.3 | 20.8 ± 4.7 | 64 ± 0 | 94.4 ± 3.2 |
| 220 | ✓ | ✓ | ✓ | Pers. | ✗ | Real | 4 | 10.4 | 37.6 ± 10.6 | 2.4 ± 3.2 | 0.8 ± 1.6 | 0 ± 0 | 0 ± 0 | 56.8 ± 6.9 |
| 220 | ✓ | ✓ | ✓ | Ortho. | ✗ | Real | 4 | 22.9 | 72.8 ± 3 | 7.2 ± 1.6 | 11.2 ± 4.7 | 0 ± 0 | 12 ± 0 | 53 ± 5.2 |

Table A2: Ablations results for RVT on RLBench with metrics for each task.