# OpenReview forum: "RVT: Robotic View Transformer for 3D Object Manipulation"
_robot-learning.org/CoRL/2023/Conference — CoRL 2023 Oral_

### Official Review · Reviewer_E55B · 2023-06-25

**Confidence:** 4
**Originality:** Good
**Technical Quality:** Excellent
**Clarity Of Presentation:** Excellent
**Impact:** 4

**Recommendation:**

Strong Accept: I recommend accepting the paper and will argue for my recommendation even if other reviewers hold a different opinion.

**Review:**

Strengths of the paper and RVT:

1. Perceiver-Actor (PerAct), a model proposed at CoRL 2022, is quickly getting a lot of traction in the robot learning community (with lots of citations for a recent paper). I view RVT as the next version of PerAct, and it has 36X faster training time and 2.6X faster inference time, for the same performance. This is a very compelling result because PerAct's training time and requirements are a huge limitation to its usage in the research community.

2. RVT can use a small number of demonstrations to train a single multi-task policy for a wide variety of language- and image-conditioned 3D manipulation tasks, where the policy has to predict full 6D robot end-effector poses, as well as a gripper opening/closing choice.

3. There are a lot of simulation experiments using RLBench and compelling real world results.

4. There are lots of ablations and analysis of the design choices for RVT.

Minor comments and questions:

1. For Figure 1, why does the performance of RVT stagnate after 60%? Would it possibly decrease if it was trained longer, due to overfitting?

2. In line 272: can the paper briefly clarify why "this will void the gains from using orthographic projections, 3D augmentation, and adding correspondences"?

There are a few limitations of the paper as mentioned in the conclusion. In addition to these, I would first suggest adding that performance is still only 60% (from Figure 1) so there's still more to go. Second, some might view this paper as "only" an engineering contribution built on top of PerAct. However, I think the usage of multiple views here results in sufficient technical novelty.


**Quality Of The Limitations Section:**

Limitations are addressed clearly

**Questions For Rebuttal:**

I don't have specific questions other than those implied by the review above. Please correct me if I happen to be mistaken in my understandings.


**Robotics Focus:**

Sufficient demonstration on hardware

**Summary Of Paper:**

This paper introduces the Robotic View Transformer (RVT) for manipulating objects in 3D. RVT leverages Behavioral Cloning to learn from a small number of demonstrations, enabling it to execute various manipulation tasks using a single policy. By utilizing a language description of the task and a set of RGB-D images as input, RVT predicts the target end-effector pose and gripper open/close state at each time step to generate actions. A notable architectural innovation of RVT is the incorporation of "multi views," where virtual cameras are strategically positioned in the scene and aggregated through an attention mechanism to make predictions in a 3D space. Comparisons with Perceiver-Actor (PerAct) from CoRL 2022 show that RVT gets similar performance but trains *much* faster and has much faster inference speed.


**Summary Of Recommendation:**

Overall, this paper makes a strong contribution to the research community. PerAct was already one of my favorite papers due to its multi-task and manipulation performance, and I'm happy to see research that makes training and running it much faster. (I am not an author on PerAct nor do I collaborate with the authors.)

---

> ### Author Response · Authors · 2023-08-08
> **Authors' First Response to Reviwer E55B**
>
> We sincerely appreciate all your suggestions and kind help in improving our paper. We are very excited that you found our paper thorough and enjoyable to read. Hopefully, the following addresses your concerns.
>
> ---
> **A1:** For Figure 1, why does the performance of RVT stagnate after 60%? Would it possibly decrease if it was trained longer, due to overfitting?
>
> **Q1:** This is a great question. As common in training transformers, we use a learning rate decay schedule. At the end of the schedule the training converges because the learning rate is low. We are currently investigating the training behavior by loading the model and training longer with larger learning rate. We will report back soon.
>
> ---
> **A2:** In line 272: can the paper briefly clarify why "this will void the gains from using orthographic projections, 3D augmentation, and adding correspondences"?
>
> **Q2:** Thanks for pointing it out. We will clarify it in the paper.
>
> Specifically, as we see in Table 2 Left, orthographic projections, 3D augmentation and adding xyz image (correspondence) improve performance. However, these could only be added after re-rendering, because:
>
> First, real-world cameras generally provide only perspective projection and not orthographic projection. To obtain orthographic projection, re-rendering is needed.
>
> Secondly, effects of 3D augmentation like rotation of the object cannot be trivially simulated in the image without re-rendering. We first create a 3D representation, apply the augmentation and re-render.
>
> Finally, adding xyz image between points in images first requires explicitly building the 3D point cloud from images, and rendering the xyz image.
>
> ---
> **A3:** There are a few limitations of the paper as mentioned in the conclusion. In addition to these, I would first suggest adding that performance is still only 60% (from Figure 1) so there's still more to go.
>
> **Q3:** Thanks for the great suggestion. We will add that there is still scope for more improvement in the limitations.
>
> ---
> Again we are hopeful that you will find our response satisfactory. We will be grateful if you can let us know if any of your concerns remain unresolved.

---

> > ### Comment · Reviewer_E55B · 2023-08-12
> > **Thank you for the response**
> >
> > Thank you for the helpful response.

---

> > > ### Author Response · Authors · 2023-08-15
> > > **Authors' Second Response to Reviwer E55B**
> > >
> > > Again, thanks for your feedback. We have now completed the new experiment. Specifically,
> > >
> > > ---
> > > **Q1:** For Figure 1, why does the performance of RVT stagnate after 60%? Would it possibly decrease if it was trained longer, due to overfitting?
> > >
> > > **A1.2 (Extending on A1):** Thanks for the great suggestion. We did this experiment now, specifically, we loaded the original trained RVT and trained it further with multiple initial learning rates ($1.2\times10^{-3}$, $0.6\times10^{-3}$) for 15 more epochs. We find that although the loss decreases further, the evaluation performance stagnates around 60% (61.4% for $1.2\times10^{-3}$; 59.2% for $0.6\times10^{-3}$)

---

### Official Review · Reviewer_CGaw · 2023-07-19

**Confidence:** 4
**Originality:** Good
**Technical Quality:** Very Good
**Clarity Of Presentation:** Good
**Impact:** 3

**Recommendation:**

Weak Accept: I recommend accepting the paper, but will not argue for my recommendation if the majority of other reviewers have a different opinion.

**Review:**

Strengths:

The paper presents an interesting method to improve the accuracy and speed of 3D object manipulation. The writing is clear and easy to read. The author also conducted a thorough literature review of related works in the field of visual object manipulation and multi-task policy learning.

Issues:

One issue in the real robot experiments in the paper is that there seems to be no quantitative comparison between the proposed method and the baseline. It would be great if the authors could consider running PerAct on real robot environments to confirm that their method still outperforms the best baseline in the sim experiments.

The paper also seems to lack discussion of the relationship between their work and other recent works that make use of transformer-based or diffusion-based architectures for performing 3D object manipulation [1, 2] (note that those works went public on Arxiv at least a month before the CoRL deadline).

Some clarification questions:
- In Section 3 Line 160, the authors mention that the heatmaps of multiple virtual views are back-projected to 3D. Could the authors elaborate on how this process is done? Is there a mapping between the 2D pixels in the virtual views and the 3D points in the point cloud? What does the algorithm do to the parts of the image where there are no corresponding points in the 3D space? Do the heatmap values from multiple views get averaged after they are back-projected onto a set of points?
- PerAct extracts a set of keyframe actions from the demo by capturing bottleneck end-effector poses in the action sequence that has (1) near-zero joint velocities or (2) an unchanged gripper open state. It seems from line 292 that the keyframe actions are specified in a different way in RVT. Could the authors clarify the difference between the two ways of collecting demos?
- In the ablation experiments, the authors argued in lines 271-274 that rearranging sensor cameras to match re-rendering views in order to bypass re-rendering would "void the gains from using orthographic projections, 3D augmentation, and adding correspondences", but did not seem to actually run such an experiment. Could the authors add such an experiment in Table 2 left? Also, it would be interesting to see ablation experiments where only the sensor camera locations change on top of the original experiment (ie. Table 2 left first row).
- In the real robot experiments, only one camera view is used. How carefully does this view have to be selected so that the method performs well? Would the performance of the method improve if more cameras are placed in the workspace?

[1] Zhao et al. Learning Fine-Grained Bimanual Manipulation with Low-Cost Hardware. RSS 2023.

[2] Chi et al. Diffusion Policy: Visuomotor Policy Learning via Action Diffusion. RSS 2023.

**Quality Of The Limitations Section:**

Limitations are addressed clearly

**Questions For Rebuttal:**

- Include a baseline for the real robot result.
- Discussions of related works in learning 3D manipulation mentioned above.
- Include explanations for the clarification questions in the paper or appendix.
- Include more details about hyperparameter choices of the method and baselines, loss function, and real robot setup in the appendix.

**Robotics Focus:**

Sufficient demonstration on hardware

**Summary Of Paper:**

This work proposes a new framework (RVT) for 3D object manipulation. Instead of using voxel-based or NeRF-based methods, the authors propose to synthesize virtual 2D images from a point cloud of the scene and feed the virtual images into a multi-view transformer. The multi-view transformer predicts a 2D heatmap for each view, which is then back-projected to the 3D space to determine the next best end-effector position. The global features aggregated from the outputs of the transformer are used to predict the end-effector rotation, gripper actions, and collision indicator.

The authors performed thorough simulation experiments on RLBench, where results show that RVT not only outperforms PerAct in 16 of the 18 RLBench tasks but also achieves much faster training time and inference speed compared to PerAct. The authors also conducted a set of real-world experiments, showing that their proposed method could reliably perform a range of 5 manipulation tasks on a real robot platform.

**Summary Of Recommendation:**

The paper introduces a novel method for 3D object manipulation that could inspire future efforts in this area. The contributions are clear and the experiments are well-executed. The paper could improve with more real robot quantitative results and some minor writing fixes. If the authors could address those concerns, I would be happy to recommend the manuscript for publication.

---

> ### Author Response · Authors · 2023-08-08
> **Authors' First Response to Reviewer CGaw**
>
> Thank you so much for taking the time to provide valuable feedback and critique on our work. We are excited that you found our work worthy of being accepted. We are hopeful that you will find our responses satisfactory.
>
> ---
> **Q1 (Issues 1):** One issue in the real robot experiments in the paper is that there seems to be no quantitative comparison between the proposed method and the baseline.
>
> **A1:** Thanks for the suggestion! Unfortunately, PerAct does not provide a pre-trained real world model that we can use. Therefore, for this experiment, we would need to train and evaluate PerAct in our setup. We are trying our best to execute this experiment within the rebuttal time.
>
> ---
> **Q2 (Issues 2):** The paper also seems to lack discussion of the relationship between their work and other recent works that make use of transformer-based or diffusion-based architectures for performing 3D object manipulation [1, 2] (note that those works went public on Arxiv at least a month before the CoRL deadline).
>
> **A2:** Thanks for bringing these papers to our attention. Although we compare with other transformer based models for object manipulation in L98-103, unfortunately, the specific papers missed our radar as they were not published at the time of the submission. We will add a discussion in a future version.
>
> ---
> **Q3 (Clarification Question 1):** In Section 3 Line 160, the authors mention that the heatmaps of multiple virtual views are back-projected to 3D. Could the authors elaborate on how this process is done? Is there a mapping between the 2D pixels in the virtual views and the 3D points in the point cloud? What does the algorithm do to the parts of the image where there are no corresponding points in the 3D space? Do the heatmap values from multiple views get averaged after they are back-projected onto a set of points?
>
> **A3:** Yes, to calculate the heatmap value at 3D location, we map 3D points to 2D pixels in the virtual views. However, we consider not just the points in the point cloud, but all points in the 3D scene bounds that are distributed at a resolution of (h x h x h), where (h x h) is the resolution of the virtual image. For each point, the heatmap value from multiple views are averaged. We will clarify it further in the paper.
>
> ---
> **Q4 (Clarification Question 2):** PerAct extracts a set of keyframe actions from the demo by capturing bottleneck end-effector poses in the action sequence that has (1) near-zero joint velocities or (2) an unchanged gripper open state. It seems from line 292 that the keyframe actions are specified in a different way in RVT. Could the authors clarify the difference between the two ways of collecting demos?
>
> A4: We follow the same pipeline as PerAct to extract keyframes from demonstrations. L292 describes our scheme for collecting target poses for real world demonstrations. The difference is that PerAct uses a VR controller to specify target poses while we do so by kinesthetically moving the robot. Once a real-world demonstration is collected, the process of extracting keyframes is the same.
>
> ---
> **Q5 (Clarification Question 3):** In the ablation experiments, the authors argued in lines 271-274 that rearranging sensor cameras to match re-rendering views in order to bypass re-rendering would "void the gains from using orthographic projections, 3D augmentation, and adding correspondences", but did not seem to actually run such an experiment. Could the authors add such an experiment in Table 2 left? Also, it would be interesting to see ablation experiments where only the sensor camera locations change on top of the original experiment (i.e. Table 2 left first row)
>
> **A5:** Thanks for the suggestion. For this experiment, we would need to first create a new image dataset beyond what is provided in PerAct. We are trying our best to complete this experiment in the rebuttal time frame.
>
> ---
> **Q6 (Clarification Question 4):** In the real robot experiments, only one camera view is used. How carefully does this view have to be selected so that the method performs well? Would the performance of the method improve if more cameras are placed in the workspace?
>
> **A6:** We used a standard third person view that is in front of the robot. We ensured that the workspace is visible in the camera but no particular effort was put in adjusting the view to make the method perform well. Potentially having more cameras could improve the method.
>
> ---
> **Q7 (Question For Rebuttal 4):** Include more details about hyperparameter choices of the method and baselines, loss function, and real robot setup in the appendix.
>
> **A7:**  We use the official code and hyperparameters for PerAct. We will release the code for our model as well as add more details in the appendix.
>
> ---
> Again we are hopeful that you will find our response satisfactory. We will report back the pending experiments as they get completed. Other than those, we will be grateful if you can let us know if any of your concerns remain unresolved.

---

> > ### Author Response · Authors · 2023-08-15
> > **Authors' Second Response to Reviewer CGaw**
> >
> > Again, thanks for your help with improving our paper. We have now completed the new experiments. Specifically,
> >
> > ---
> > **Q1 (Issues 1):** One issue in the real robot experiments in the paper is that there seems to be no quantitative comparison between the proposed method and the baseline.
> >
> > **A1.2 (Extending on A1):** Thanks for the suggestion. As noted in our earlier response, PerAct does not provide a pre-trained real-world model that we can use. Therefore, we now trained and evaluated the official PerAct model by training on the same real-world dataset we used for training RVT. We evaluated PerAct in the two settings: (1) where it was trained for the same number of steps as RVT (\~16k, batch size 24), and (2) where it was trained for 5X more steps (\~80k) than RVT to ensure convergence. We have summarized the success rates (with and without tasks with marker objects, Sec. 4.2) below. Similar to our findings in simulation, we find that RVT outperforms PerAct in the real world.
> >
> > | Task                    | PerAct (16k) |PerAct (16k) | PerAct (80k)| PerAct (80k) | RVT| RVT |
> > |---------------------------------|---------------|--------------|----------------|--------------|--------------|------------|
> > |			     | (+ marker) | (-marker) | (+ marker) | (-marker) |(+ marker)|(-marker)|
> > | Stack blocks                  |           30%|          30%|           50% |         50%|        100%|     100%|
> > | Press sanitizer               |           20%|          20%|          40%  |         40%|         80% |       80%|
> > | Put marker in mug/bowl |           20%|            -    |              0%|              - |         0%   |          -   |
> > | Put object in drawer       |           20%|          50%|            20%|         50%|        50% |     100%|
> > | Put object in shelf          |           50%|          50%|             30%|         30%|        50%|        50%|
> > | All tasks                         |           28%|        37.5%|            28%|      42.5%|       56% |      82.5%|
> >
> > ---
> > **Q5 (Clarification Question 3):** In the ablation experiments, the authors argued in lines 271-274 that rearranging sensor cameras to match re-rendering views in order to bypass re-rendering would "void the gains from using orthographic projections, 3D augmentation, and adding correspondences", but did not seem to actually run such an experiment. Could the authors add such an experiment in Table 2 left? Also, it would be interesting to see ablation experiments where only the sensor camera locations change on top of the original experiment (i.e. Table 2 left first row)
> >
> > **A5.2 (Extending on A5):** We now created a new image dataset with real sensors arranged in orthogonal configuration. As you suggested, with this new image dataset, we did two experiments, one directly using it and one with our pipeline (re-rendering with orthographic projection, 3D augmentation and correspondences). We find that results on this new dataset are consistent with the results on the dataset provided by PerAct, where our pipeline works better (60.0% vs 27.2%). We will add it to the paper.

---

### Official Review · Reviewer_sWvB · 2023-07-20

**Confidence:** 4
**Originality:** Good
**Technical Quality:** Good
**Clarity Of Presentation:** Very Good
**Impact:** 3

**Recommendation:**

Weak Accept: I recommend accepting the paper, but will not argue for my recommendation if the majority of other reviewers have a different opinion.

**Review:**

Strengths:
1. The paper is well-organized and easy to follow.
2. The proposed Robotic View Transformer framework achieves significant improvements compared to other baselines
3. The authors applied the proposed pipeline in the real robot experiments

Weakness:
1.  Can these multi-view images be directly achieved using multiple cameras, since the pipeline requires first reconstructing a point cloud of the scene? If real multiple-camera images are fed into the pipeline, will it improve the performance?
2. How accurate are the re-rendered images? Does the quality of these re-rendered images affect the performance of the action prediction?
3. What would happen if the predicted key frame(gripper poses) is not achievable due to collision or joint limit?



**Quality Of The Limitations Section:**

Additional details required

**Questions For Rebuttal:**

See weakness

**Robotics Focus:**

Sufficient demonstration on hardware

**Summary Of Paper:**

This manuscript introduces a multi-view transformer for 3D object manipulation.  The transformer contains an attention mechanism to gather information from multiple view images (re-rendering from virtual views) and output the target end-effort pose. The low-level controller or motion planner generates actions to achieve these target poses. The authors applied the proposed pipeline in the RLBench to demonstrate its effectiveness and also tested it in real-robot experiments.

**Summary Of Recommendation:**

The authors proposed a Robotic View Transformer framework for 3D object manipulation achieving significant improvements compared to other baselines. However, this pipeline requires first reconstructing a point cloud. How the quality of the reconstructed point cloud affects the performance is not clear to the reviewer. If multiple images are needed to reconstruct the point cloud, what are the advantages of re-rendering and how the quality of the rendered images affects the pipeline performance is not clear.

---

> ### Author Response · Authors · 2023-08-08
> **Authors' First Response to Reviewer sWvB**
>
> Thanks a lot for providing your valuable feedback on our work. We are thrilled that you consider our work deserving of acceptance. We are hopeful that our response will address your concerns.
>
> ---
> **Q1 (Summary Of Recommendation):** The authors proposed a Robotic View Transformer framework for 3D object manipulation achieving significant improvements compared to other baselines. However, this pipeline requires first reconstructing a point cloud. How the quality of the reconstructed point cloud affects the performance is not clear to the reviewer.
>
> **A1:** We would like to clarify a potential confusion. In RVT, the point cloud is computed from the RGB-D input. This step of recovering point cloud from depth is the same as the state-of-the-art baseline, PerAct. We do not assume any extra reconstruction steps (like SLAM or NeRF) beyond what is used in PerAct.
>
> Thanks for the suggestion to investigate how the quality of point clouds affect performance. Note that prior methods like PerAct would also have the same issue. Still, to further investigate this in RVT, we are conducting an experiment in simulation with varying noise in the input depth. We will report back when the results are ready.
>
> ---
> **Q2 (Summary Of Recommendation):** If multiple images are needed to reconstruct the point cloud, what are the advantages of re-rendering?
>
> **A2:** We want to clarify that RVT does not necessarily need multiple images. As done in our real world experiments, RVT can work even with a single RGB-D camera.
>
> Orthogonal to the number of cameras, the use of re-rendering can provide several advantages such as re-anchoring to task advantageous view, 3D augmentation, and altering the projection model, as mentioned in L57-60 and L272-273.
>
> ---
> **Q3 (Weakness 1):** Can these multi-view images be directly achieved using multiple cameras, since the pipeline requires first reconstructing a point cloud of the scene?
>
> **A3:** Yes, the multi-view images can be directly achieved using multiple cameras. However, there are two sets of disadvantages to this:
>
>
> - As noted in the paper (L273-274), a multi-camera setup like ours with 5 orthogonal cameras (Fig 3.) is difficult to maintain. In contrast, our current system with a single RGBD sensor is cheaper and easier to deploy.
>
> - As noted in the paper (L244-245, L252-L254, L255-256), directly using multi-view images without any re-rendering would prevent us from using orthographic projection, 3D augmentation and point correspondence. All these significantly boost performance (Table 2 Left) but require re-rendering.
>
> ---
> **Q4 (Weakness 1):** If real multiple-camera images are fed into the pipeline, will it improve the performance?
>
> **A4:** We compared re-rendering at orthogonal locations vs directly using multiview images from original RLBench camera locations (Fig. 3d) and found that re-rendering works significantly better (Table 2 Left). As discussed in A3, this is because re-rendering allows us to use orthographic projection, 3D augmentation and point correspondence (see Sec 4.1 Ablation Study).
>
> For the additional study, where we directly use camera images at the same location as our virtual camera, we would need to first create a new image dataset beyond what is provided in PerAct. We are trying our best to complete this experiment in the rebuttal time frame.
>
> ---
> **Q5 (Weakness 2):** How accurate are the re-rendered images? Does the quality of these re-rendered images affect the performance of the action prediction?
>
> **A5:** The accuracy and quality of the re-rendered images depends on the quality of the depth from the RGB-D sensor. We find the depth from the Azure Kinect sensor to be sufficiently good for our real world experiments.
>
> As discussed in A1, to quantify how the quality of re-rendered images might affect performance, we are conducting an experiment in simulation with varying noise in the input depth. We will report back when the results are ready.
>
> ---
> **Q6 (Weakness 3):** What would happen if the predicted key frame (gripper poses) is not achievable due to collision or joint limit?
>
> **A6:** If the predicted gripper pose is not achievable, then the episode will fail. When evaluating our method, we mark such episodes as being unsuccessful in sim and real.
>
> ---
> Again we are hopeful that our response addresses your concern. We will report back the pending experiments as they get completed. Apart from them, please let us know if any concerns still remain.

---

> > ### Author Response · Authors · 2023-08-15
> > **Authors' Second Response to Reviewer sWvB**
> >
> > Thanks for your valuable feedback. We have now completed the new experiments. Specifically,
> >
> > ---
> > **Q1 (Summary Of Recommendation):** The authors proposed a Robotic View Transformer framework for 3D object manipulation achieving significant improvements compared to other baselines. However, this pipeline requires first reconstructing a point cloud. How the quality of the reconstructed point cloud affects the performance is not clear to the reviewer.
> >
> > **A1.2 (Extending on A1):** As mentioned in our previous response (A1), we want to clarify that RVT and PerAct (primary baseline) have the same pipeline to recover point clouds from single or multiple RGB-D images; and that RVT does not assume any additional reconstruction steps like SLAM or NeRF.
> >
> > Nevertheless, to further investigate how the quality of the point cloud affects performance in RVT, we do additional experiments in simulation. Specifically, we add Gaussian noise with varying standard deviation (2.5mm, 5mm, 1cm, 2cm, and 4cm) to the original point cloud. We add this noise both during training and evaluation to simulate sensor noise in both phases.
> > The success rate is 62.9 for no noise, 62.0 for 2.5mm, 61.6 for 5mm, 56.4 for 1cm, 58.7 for 2cm and 51.7 for 4cm standard deviation noise. We find that RVT is robust to 2cm standard deviation noise in the point cloud and its performance degrades gracefully with more noise. For reference, the depth measurements in Intel RealSense D400 camera has an error of 2.5mm to 5mm for an object at 1m from the camera (source: https://www.intel.com/content/www/us/en/support/articles/000026260/emerging-technologies/intel-realsense-technology.html)
> >
> > ---
> > **Q4 (Weakness 1):** If real multiple-camera images are fed into the pipeline, will it improve the performance?
> >
> > **A4.2 (Extending on A4):** As we noted in A4, we previously compared (Table 2 Left) our pipeline with re-rendering against directly using multiview images provided in PerAct, and found that our pipeline works better.
> > As suggested, we conducted a new experiment, where we created a new image dataset from RLBench, with real perspective cameras placed at orthogonal locations. We trained two models on this dataset, one directly using the provided images and one with our pipeline. We find that our pipeline performs better (60.0% vs 27.2% success rate) because it allows for orthographic projection, 3D augmentation, and point correspondence (Table 2 Left).

---

### Author Response · Authors · 2023-08-15
**Authors' Common Response**

We sincerely appreciate all the feedback and the help with improving our paper. We are excited that all the reviewers appreciated our work. Specifically, they found that our work “makes a strong contribution to the research community” (E55B) and “could inspire future efforts in this area” (CGaw). Reviewers commented that our proposed method RVT is technically “novel” (CGaw, E55B), “interesting” (CGaw), “achieves significant improvements compared to other baselines” (sWvB) and “trains much faster and has much faster inference speed” (E55B). They also liked that we performed “thorough simulation experiments” (CGaw) with “lots of ablations and analysis of the design choices” (E55B) and also presented “compelling real world results” (E55B). Lastly, all the reviewers found our paper to be well-organized and easy to read.

We have responded to the concerns of the reviewers individually. We hope our earlier response was satisfactory. We have also finished all additional experiments and reported their results.

---

### Decision · Program_Chairs · 2023-08-30

**Decision:**

Accept (Oral)

**Comment:**

The paper presents an extension of PerAct using a continuous representation introducing a multi-view transformer architecture that achieves significant improvements over the prior method, and the paper also provides real world evaluation of the method. Overall, they idea of keyframe prediction of end-effector poses for complex manipulation tasks is a method that  gains a lot of traction.

The reviewers have evaluated mostly positive the work, and the authors have done a good job in the rebuttal period, particularly providing additional results on the real world comparison of RVT and PerAct, showcasing the superior performance of PerAct. The addition of clarifications on the experimental process and the the additional real-world comparison are essential for the paper and I expect to see them added in the camera-ready version of the paper.